# Predictive Markers of Immunogenicity and Efficacy for Human Vaccines

**DOI:** 10.3390/vaccines9060579

**Published:** 2021-06-01

**Authors:** Matthieu Van Tilbeurgh, Katia Lemdani, Anne-Sophie Beignon, Catherine Chapon, Nicolas Tchitchek, Lina Cheraitia, Ernesto Marcos Lopez, Quentin Pascal, Roger Le Grand, Pauline Maisonnasse, Caroline Manet

**Affiliations:** 1Immunology of Viral Infections and Autoimmune Diseases (IMVA), IDMIT Department, Institut de Biologie François-Jacob (IBJF), University Paris-Sud—INSERM U1184, CEA, 92265 Fontenay-Aux-Roses, France; matthieu.van-tilbeurgh@cea.fr (M.V.T.); lemdani.katia@gmail.com (K.L.); anne-sophie.beignon@cea.fr (A.-S.B.); catherine.chapon@cea.fr (C.C.); lina.cheraitia@yahoo.com (L.C.); ernesto.marcos-lopez@cea.fr (E.M.L.); quentin.pascal@cea.fr (Q.P.); roger.le-grand@cea.fr (R.L.G.); pauline.maisonnasse@cea.fr (P.M.); 2Unité de Recherche i3, Inserm UMR-S 959, Bâtiment CERVI, Hôpital de la Pitié-Salpêtrière, 75013 Paris, France; nicolas.tchitchek@sorbonne-universite.fr

**Keywords:** vaccines, systems immunology, predictive biomarkers, vaccine signatures, preclinical models, high-throughput technologies, in vivo imaging, unsupervised analyses, machine learning

## Abstract

Vaccines represent one of the major advances of modern medicine. Despite the many successes of vaccination, continuous efforts to design new vaccines are needed to fight “old” pandemics, such as tuberculosis and malaria, as well as emerging pathogens, such as Zika virus and severe acute respiratory syndrome coronavirus 2 (SARS-CoV-2). Vaccination aims at reaching sterilizing immunity, however assessing vaccine efficacy is still challenging and underscores the need for a better understanding of immune protective responses. Identifying reliable predictive markers of immunogenicity can help to select and develop promising vaccine candidates during early preclinical studies and can lead to improved, personalized, vaccination strategies. A systems biology approach is increasingly being adopted to address these major challenges using multiple high-dimensional technologies combined with in silico models. Although the goal is to develop predictive models of vaccine efficacy in humans, applying this approach to animal models empowers basic and translational vaccine research. In this review, we provide an overview of vaccine immune signatures in preclinical models, as well as in target human populations. We also discuss high-throughput technologies used to probe vaccine-induced responses, along with data analysis and computational methodologies applied to the predictive modeling of vaccine efficacy.

## 1. Introduction

Vaccines are the most effective preventive measure ever developed in the fight against diseases. They have led to the eradication of smallpox and to a major reduction in the incidence of diseases such as diphtheria, tetanus or poliomyelitis. Nevertheless, the need for new vaccines has never been so critical as demonstrated by the recent SARS-CoV-2 pandemic. Novel vaccines are also required to fight against “old” diseases like malaria and tuberculosis [1], which are still responsible for millions of new infections and hundreds of thousands of deaths each year [2]. Improving existing vaccines is also important to increase disease control and prevent outbreaks of re-emerging pathogens [3]. For example, despite its high efficacy, the live-attenuated yellow fever (YF) vaccine cannot be safely administrated to immunocompromised individuals, and its slow production can lead to vaccine shortage and subsequent inadequate control of YF epidemics [4].

One of the main goals in vaccinology is to identify factors that reflect vaccine-induced immune responses and thus provide biomarkers of vaccine immunogenicity and efficacy. A biomarker can be defined as “a characteristic that is objectively measured and evaluated as an indicator of normal biological processes, pathogenic processes, or pharmacological responses to a therapeutic intervention” [5]. By extension, a vaccine signature can be defined as a set of biomarkers that statistically differ between vaccinated and non-vaccinated individuals and are indicative of vaccine-induced biological responses.

The emerging field of systems vaccinology aims to identify biomarkers and immune signatures that correlate with vaccine efficacy to decipher protective immune mechanisms. A correlate of protection is defined as a biomarker or an immune mechanism that is “statistically related to and responsible for protection” [6,7]. Thus, the possibility to predict vaccine efficacy is tightly intertwined with the notion of correlates of protection which can be characterized by various types of biomarkers.

High-throughput technologies have rapidly expanded over the last several years and have been frequently employed in systems vaccinology studies, making it possible to extend the range of biomarkers included in vaccine signatures [8,9,10,11,12]. New approaches in data analysis methodologies and computational modeling take vaccine signatures a step further by giving rise to the possibility of identifying immune responses that correlate and/or predict vaccine efficacy.

Although systems vaccinology ultimately aims to develop predictive models of vaccine efficacy in human populations, applying the same approach to animal models, which allow the use of a wide range of tools in controlled study designs, empowers vaccine research and improves preclinical studies. Notably, high-throughput imaging technologies can be used in preclinical models to characterize immune responses at the whole-body and tissue levels [13,14] and to define more comprehensive vaccine signatures. In addition, using systems vaccinology in models such as non-human primates (NHPs), which are highly predictive of the human immune and vaccine responses, will increase the translation of discoveries from animal studies to human clinics.

Here, we review approaches to identify biomarkers and signatures of vaccine responses in preclinical models and humans. We provide an overview of high-throughput technologies used to probe vaccine-induced responses, including in vivo and in vivo imaging technologies. We also present data analysis and computational methodologies used to define signatures that correlate with and potentially predict vaccine efficacy.

## 2. Identification of Biomarkers and Signatures of Vaccine Responses

One of the main goals in systems vaccinology is to identify a strong and reliable vaccine signature that statistically differs between immunized and non-immunized individuals that can be easily measured in the blood and at a reasonable cost.

However, protection conferred by vaccination results from complex interactions between innate and adaptive immunity and there are considerable differences between individuals in the response to immunization. Such variation, mediated by both host factors and vaccine properties, precludes the description of a universal marker of vaccine efficacy (Figure 1).

Indeed, many studies have demonstrated the effects of sex on vaccine responses [15,16,17], as well as the preexistent immunological background and non-immunological co-factors. Host genetic background also modulates immune responses to vaccines, for example, several studies [8,18] demonstrated that different signatures of the yellow fever 17D strain (YF-17D) vaccine can be found between human cohorts and Pogorelyy et al. [11] even found differences between monozygotic twins. In the past years, the influence of host genetic factors has been investigated more precisely through vaccinomics [19,20]. Genome wide association studies (GWAS) have identified several polymorphisms in the human leukocyte antigen (HLA) gene associated to a poor or non-response to the hepatitis B virus (HBV) [21,22,23,24] and to the measles, mumps, and rubella (MMR) [25] (REF) vaccines. Other genes encoding various cytokines, Toll-like-receptors (TLR) and their signaling molecules have been associated to increased or decreased HBV and MMR vaccine efficacy, as reviewed extensively by Omersel and Kuzelicki [19].

Similarly, vaccine composition influences the dynamics of immune parameters, such as immune-cell migration to the injection site or the immune-cell transcriptional profile [26,27]. In addition, Li et al. demonstrated that the vaccine signature differs depending on the type of immunogen, similarly to pathogens targeting cells and tissues through various mechanisms [27].

Consequently, host- and vaccine-related factors, extensively reviewed by Zimmermann and Curtis [28], may influence and shape biomarker expression by affecting immune responses and, subsequently, vaccine signatures.

The characterization of extensive vaccine signatures is further influenced by the different types and sources of biomarkers, as discussed hereafter.

### 2.1. What Types of Biomarkers Can Be Used to Define Vaccine Signatures?

Antibody responses are widely used to assess vaccine responses [29]. However, they may not always represent the best biomarkers of vaccine efficacy, especially when vaccine-induced protection is mediated by cellular immunity. In addition, effective neutralizing antibody responses may take months or years to be induced, such as for broadly human immunodeficiency virus (HIV)-neutralizing antibodies, and can jeopardize the use of antibodies as early biomarkers of vaccine-induced protection [30]. Moreover, effective antibody responses can also be induced through T-cell independent pathways, suggesting that unconventional or unknown biomarkers could correlate with and predict such responses [31]. Indeed, Chaudhury et al. showed that the sole use of such analysis is too reductive to identify differences between immune responses. Indeed, their predictive model of the malaria vaccine-induced immune response also integrates other immune parameters, such as IL-4 and IL-6 levels, which are variables of importance in different tissues such as blood or liver [32].

Years of immune system screening have demonstrated that immune responses to pathogens and vaccines are highly multifactorial and involve numerous diverse actors. Historically, the quality of the vaccine response has been closely related to adaptive lymphoid cell populations, including effector responses of CD8^+^ T cells and CD4^+^ T helper cells, as well as immune memory. However, the role of innate cell populations has been recently re-evaluated to consider their influence in initiating and orienting the adaptive response. Consequently, innate cells are increasingly being studied as early biomarkers of vaccine efficacy. Notably, several studies have shown that innate myeloid cells, such as neutrophils, monocytes, and innate lymphoid cells (ILCs), such as natural killer (NK) cells, are of interest in defining vaccine signatures [33,34,35,36]. The diverse innate and adaptive immune-cell subsets, including unconventional subsets such as Tγδ lymphocytes and ILCs, represent a large source of potential biomarkers that could be used to define signatures of vaccine response. Furthermore, it has also been shown that the route of immunization can orient vaccine responses, suggesting that non-immunological cells within target tissues could broaden the list of potential biomarkers of vaccine responses [26,37].

Until now, systems vaccinology studies have mainly used transcriptomic techniques to investigate vaccine responses and identify biomarkers to define vaccine signatures [8,11,27]. However, the multitude of chain reactions induced by vaccine injection may affect cellular activity at different levels, such as epigenetic modifications, protein levels, or enzyme activity, which may also constitute a large reservoir of potential biomarkers [6,38]. Other factors, such as cytokines or growth factors should not be excluded from the list of candidate biomarkers of vaccine responses [32,39]. Finally, variations in the composition of the gut microbiota have been shown to influence vaccine-induced responses, thus further increasing the spectrum of possible immune biomarkers [40].

Thus, a biomarker can be based on the detection of gene transcripts, proteins, and metabolites at the single cell or cell population level. Furthermore, data provided by histology, tissue imaging, and even clinical metadata are still poorly represented in the emerging field of systems vaccinology but would also empower holistic approaches aiming to define comprehensive signatures of vaccine efficacy (Figure 2).

### 2.2. From What Samples Can We Identify Vaccine Response Biomarkers?

In terms of where to look, the search for immune biomarkers is not only oriented by the factors to be identified per se and existing knowledge on the immunological processes involved but is also further constrained by practical aspects, including technical feasibility, as well as ethical and financial restrictions. Given such limitations, **blood** constitutes the main source of biomarkers, as it is the most accessible sample in humans, allowing the study of circulating cells, soluble factors (plasma), and antibodies (serum). The diversity and functionality of circulating immune cells have been extensively investigated, especially since the rise of high-throughput technologies [46,47], providing a good snapshot of the global immune state. Furthermore, peripheral blood mononuclear cells (PBMCs) may also reflect immune activity in other tissues, as shown by DeGottardi et al. for circulatory CXCR5^+^ CD4^+^ T cells and lymph node T follicular helper (TFH) activity [48]. Others have also used blood samples to identify correlates of protection for several vaccines, such as cell-mediated immunity for varicella-zoster virus (VZV) or humoral response for smallpox [49,50,51]. Finally, predictive models have also been developed from PBMC samples, such as that described by Querec et al., predicting the YF-17D CD8^+^ T cell response with high accuracy [52].

However, immune responses imply cell mobilization and maturation processes within various tissues, including lymphoid (lymph nodes, bone marrow) and non-lymphoid organs (liver, skin, muscles, etc.). Contrary to peripheral blood, other human tissues are not easy to access without invasive procedures or advanced imaging techniques and have rarely been included in large-scale studies (see Section 4). However, certain tissues may be accessible by performing small biopsies, such as fine-needle aspiration, commonly used for tumor biopsies or skin explants [53], although the size of such samples may limit the use of techniques that require large numbers of cells. On the other hand, preclinical models allow much wider access to all organs and thus the identification of relevant biomarkers that reflect tissue-based immune mechanisms.

Moreover, differences in immune-cell colonization between tissues have been demonstrated and must be considered when searching for vaccine biomarkers [37,46]. Indeed, the injection site may be used to identify early biomarkers, such as antigen uptake [42,45], whereas biomarkers of immune memory can be detected in lymph nodes and/or bone marrow [54,55]. Deep characterization of vaccine responses may also need to account for pathogen tropism for the identification of biomarkers. For example, liver or spleen could be a source of biomarkers in the case of YF-17D vaccination [56].

Furthermore, mucosal immunity can be crucial in the protection conferred by certain vaccines. Darrah et al. demonstrated that intravenous immunization of Rhesus macaques with bacille Calmette-Guérin (BCG) vaccine (Danish Strain 1331, Statens Serum Institute, Copenhagen, Denmark) induced a higher CD4^+^ and CD8^+^ T cell responses in cells from bronchoalveolar lavages than by other immunization routes [57]. Pattyn et al. reviewed studies in which human papillomavirus (HPV)-specific antibodies were measured in cervicovaginal secretions in response to HPV vaccine [58], which implies that, similarly to tissues, biological fluids may also represent a potential source of biomarkers.

In conclusion, various types of biological samples reflect different aspects of the immune response and influence the biomarkers that can be identified. Thus, investigating diverse types of samples leads to deeper characterization of vaccine responses and allows unravelling of the vaccine biomarkers involved in various immune processes.

### 2.3. At What Time Should We Identify Vaccine Response Biomarkers?

Temporality is another important dimension to consider when identifying biomarkers to define vaccine signatures, as they would vary in parallel with the stages of the immune response, from the baseline to effector and memory stages.

Recently, a study aiming to better understand the impact of early innate parameters on the adaptive response against modified vaccinia Ankara (MVA) identified cellular and molecular events specifically activated by MVA immunization as early as six hours post-vaccination [36]. Other studies have also shown that early time points, within 24 h post-immunization, are of interest, as early innate biomarkers correlated with the adaptive responses to the vaccine [8,33]. Furthermore, innate immune biomarkers can be detected as late as two months after vaccination, as observed for granulocytes, monocytes, and dendritic cells (DCs) from NHPs immunized with MVA [35]. Apart from innate immunity, certain biomarkers may need weeks to appear, in particular those associated with adaptive responses and maturation processes that lead to vaccine memory. In addition, the identification of immune parameters that correlate with long-term memory induced by vaccines may require searching for biomarkers several months after immunization. For example, Bhaumik et al. investigated the long-term persistence of immune memory following one- or two-dose inactivated poliovirus vaccine schedules. They showed that memory B cells can be induced by both vaccine regimens, although this cell subset declined by five months after a single immunization, whereas it persisted for more than one year with the two- dose strategy [59]. Additionally, identifying predictive biomarkers at baseline, i.e., before vaccination, has drawn interest from researchers who reported proof-of-concept results on influenza [60,61], yellow fever [62], and hepatitis B [63] vaccines. Thus, the identification of biomarkers will be highly dependent on the time of sampling.

However, sampling is not the only temporal component that influences vaccine signatures. Indeed, several studies have highlighted the impact of the vaccine schedule on the distribution of cell populations or antibody production [33,64]. Thus, differential post-prime and post-boost vaccine signatures could be identified and characterized, which could provide useful insights to the design of new vaccination strategies for human populations.

Temporal considerations thus represent an important source of biomarker variation that need to be thoroughly investigated to fully understand the impact of time on the immune responses and biomarker dynamics following vaccine injection.

Beyond considerations of nature, distribution and time, the identification of vaccine signatures is also conditioned by the technologies used to measure immune-related biomarkers, as discussed in the following sections.

## 3. Conventional High-Throughput Technologies to Assess Vaccine Responses

### 3.1. High-Dimensional Flow and Mass Cytometry

Technologies for single cell analysis have become crucial in the field of vaccinology. The advances in cytometric technologies over the last several years have allowed researchers to obtain a comprehensive understanding of heterogeneity among immune cells, cell function, cellular differentiation, and signaling pathways [65] and to apply this knowledge to the discovery of biomarkers of vaccine responses [66].

Traditional fluorescent flow cytometry relies on fluorescent markers as a reporter. The emission spectra overlap of the various fluorophores, auto-fluorescence, and compensation-related issues limit the number of markers that can be simultaneously measured. On the other hand, spectral flow cytometry is based on many of the fundamental aspects of conventional flow cytometry but has unique optical collection and analytical capabilities, doubling the number of markers that can be simultaneously measured [67].

A format for flow cytometry has been developed that takes advantage of the precision of mass spectrometry. This fusion of the two technologies, called mass cytometry, enables the simultaneous measurement of up to 50 cellular features at single-cell resolution, significantly augmenting the ability of cytometry to evaluate complex cellular systems and processes [68,69]. These characteristics enable the investigation of complex and coordinated cellular systems by observing the diversity of cellular phenotypes and behaviors in a single sample [68]. This technology opens new possibilities in vaccinology, providing a tool capable of simultaneously capturing diverse aspects of cellular behavior in millions of individual immune cells.

Indeed, mass cytometry has since shown that there are hundreds of phenotypically distinct cell types in the peripheral blood of humans and animal models [36,70,71]. The ability to discriminate between these cell types is crucial to our understanding of cellular immunity and vaccine responses, and mass cytometry has become a powerful tool for this purpose. This can be illustrated by recent human and preclinical vaccine studies, which provide evidence of the phenotypic diversity of both lymphoid and myeloid cells [46].

For example, Palgen et al. characterized qualitative and quantitative differences in the recruitment of innate myeloid cells following MVA prime-boost immunization of NHPs [35]. Moreover, longitudinal mass cytometry analysis of NK cells after MVA vaccination revealed key features of cell phenotype, suggesting that the innate response to the boost is more highly coordinated between NK cells and innate myeloid cells than the response to the prime [34]. Another study highlighted the phenotypic heterogeneity of vaccine-altered circulating B cells of NHPs immunized twice with MVA, two months apart [72].

Mass cytometry has also been used to identify novel influenza vaccine-specific CD4^+^ T-cell subsets in humans [73]. In this study, the authors identified two cell clusters that responded either to influenza peptide stimulation or influenza vaccination. One cluster corresponded to pre-existing influenza virus-specific cells that presumably persisted from previous vaccination(s) or infection(s), whereas the second cluster reflected CD4^+^ T cells responding to influenza vaccination but not the specific peptides used for stimulation. Both clusters appeared to be effector memory subsets with low CCR7 and CD45RA expression and their cytokine expression profiles were distinct, as the first showed high IL-2, TNF-α, and IFN-γ expression, whereas the second mainly expressed IL-17. These detailed analyses underscore the role of CD4^+^ memory T-cell subsets in influenza virus infection and highlight the huge potential of mass cytometry to distinguish and characterize very specific cell subsets [73].

In a unique study, the ontogeny of various subsets of YF-17D-specific circulatory CXCR5^+^ CD4^+^ T cells was accessed by unsupervised analysis of mass cytometry data. The authors observed that YF virus-specific CXCR5^+^ T cells existed in multiple phenotypic clusters and that one key population was mainly ICOS^+^ PD1^+^ CD38^+^. This population most resembled germinal center T follicular helper (GC-TFH) cells, based on surface marker expression, and exhibited delayed accumulation in the periphery, implying that these T cells could be emigrants from lymph node germinal centers (GCs). The relative kinetics of their emergence following vaccination suggests that these triple positive CXCR5^+^ cells transition to become CD38^+^ ICOS^−^PD1^+^ and then CD38^−^ ICOS^−^PD1^+^ cells before accumulating in the periphery as CD38^−^ ICOS^−^PD1^−^CCR7^+^ cells. Overall, these results imply that most antigen-specific CXCR5^+^ T cells are derived from pre-TFH, and/or TFH cells [48].

Mass cytometry was also applied to comprehensively characterize the circulating immune-cell populations in elderly individuals, both before and after administration of an investigational adjuvanted protein vaccine against respiratory syncytial virus (RSV) in a Phase 1a trial. Here, mass cytometry was used to characterize the cellular response profile of enzyme-linked immunospot (ELISPOT) responders and non-responders. Principal component analysis revealed baseline differences in activated (HLA-DR^+^) CD4^+^ and CD8^+^ T cells, which were more numerous in non-responders than responders. Higher expression of HLA-DR, CCR7, CD127, and CD69 was also found in non-responders than responders using a viSNE algorithm to analyze RSV-responsive CD4^+^ and CD8^+^ T cells [74].

These studies demonstrate the potential of mass cytometry as a powerful technology to enable comprehensive profiling of immune components, thus allowing the prediction of responses to vaccines.

### 3.2. Cytokine Profiling

Immune-based assays are widely used to assess vaccine responses and since the rise of systems vaccinology, numerous studies have shown correlations between transcriptomic and cytokine signatures to various vaccines. Various technologies are currently available, from single-plex to multiplex analysis, allowing the identification of soluble vaccine biomarkers. Some have been reviewed by D. Furman and MM. Davis [75].

Most immunoassays are based on enzyme-linked immunosorbent assays (ELISAs), widely used for the robust and reliable detection of soluble components at low concentration (ng to pg/mL) by the measurement of absorbance. ELISPOT and fluoro-immunospot (FLUOROSPOT) are very sensitive ELISA-derived techniques designed to study the cytokine production capacity of cells upon specific stimulation. Contrary to ELISA, FLUOROSPOT allows the identification of poly-secreting cells by the simultaneous detection of up to four analytes using fluorescent antibodies. Despite their sensitivity, these techniques are restricted to a small number of analytes and multiplying tests to increase soluble factor detection requires larger sample volumes and is time consuming.

Therefore, multiplex immunoassays have been developed that allow rapid simultaneous quantification of a wide diversity of soluble proteins combined with high sensitivity (down to pg/mL and even fg/mL for SIMOA^®^, Quanterix, Billerica, MA, USA) with very small biological samples (down to 1 µL). ELISA-based multiplexing technologies, such as Quantibody^®^ microarrays (MSD) or xMAP^®^ Technology (Luminex^®^, Austin, TX, USA), are commonly used for cytokine profiling and immunological studies. For example, multiplex profiling of 24 cytokines/chemokines was performed on healthy individuals and tuberculosis patients and identified seven differentially expressed biomarkers between the two groups [76]. Such techniques might also reveal biomarkers in vaccine studies. These techniques rely on capture antibodies coated on slides with a spatial specificity (MSD, Quantibody^®^ microarrays) or on microbeads in suspension (xMAP^®^ Technology, Luminex^®^ assays), with a detection method related to flow cytometry. Another high-throughput multiplexing immunoassay with a sensitive detection method, based on quantitative polymerase chain reaction (qPCR) amplification of DNA oligonucleotides coupled to antibodies, has been developed: the proximity extension assay (PEA, Olink^®^, Uppsala, Sweden). PEA has demonstrated a robust ability in proteomic profiling in various diseases, including SARS-CoV-2 infection [77], and is yet another powerful tool to explore biomarkers of vaccine responses.

Cytokine measurement can also be performed using flow or mass-cytometry by intracellular cytokine staining (ICS). This method allows characterization of the cell secretion profile after their potential stimulation, along with their surface phenotype. However, despite precise identification of the producing cells, the sensitivity of ICS is generally lower than that of other immunoassays [78].

Systems vaccinology studies have embedded multiplexing technologies in the identification of soluble biomarkers of vaccine responses. Huttner et al. defined an Ebola vaccine (recombinant vesicular stomatitis virus-vectored Zaire Ebola vaccine, rVSV-ZEBOV) cytokine signature associated with biological and clinical outcomes [79]. An early IP-10 signature correlating with the antibody response was also identified after immunization with rVSV-ZEBOV [12]. More recently, several soluble factors, including IP-10, were identified using a 27-plex assay and shown to be important biomarkers of immune responses to tularemia vaccines [39].

Large-scale profiling techniques of soluble factors are thus appropriate for the deep characterization of cell secretion in response to immunogens.

### 3.3. OMICS Technologies

In addition to immune-cell phenotyping using multiparameter cytometry (flow or mass) and immunoproteomics, gene expression is widely used to identify signatures and predictors of vaccine-induced specific antibody and T-cell responses. Furthermore, transcriptomic and genomic data (using next-generation sequencing) allow to capture the diversity of the immune repertoire induced by vaccines [80], with a comprehensive quantification of full-length T cell receptor (TCR) and B cell receptor (BCR) variable region sequences [81]. Additional layers of information, including epigenomic (using ChIP-Seq or ATAC-Seq), metabolomic (using nuclear magnetic resonance spectroscopy and mass spectrometry), and that from the microbiome (using 16S rRNA or shotgun metagenomic sequencing) can be used to further characterize vaccine responses, as appropriate technologies are becoming available [38,82]. 

Transcriptomics (using RNA microarrays or RNA-Seq) analyzes sets of RNA transcripts. Microarrays are based on a fixed probe technology, whereas RNA-Seq quantifies the abundance of all transcripts (any sequence). Epigenomics studies the set of chemical modifications of the DNA and histone proteins, of which the type and position determine the chromatin structure and accessibility to the transcriptional machinery. These are key for the regulation of gene expression. Metabolomics studies the set of metabolites present in biofluids, cells, and tissues. Metabolites can be the substrates or products of metabolism. They can also originate from microorganisms, xenobiotic exposure, or the diet. The microbiome refers to the genomes of all microorganisms in our body (gut, skin, lungs, and other epithelial surfaces). The microbiota modulates host immune responses locally and globally. 

Transcriptomes and chromatin states can also be studied at the single-cell level and even simultaneously. Bulk measurements, such as whole blood or PBMC RNA-Seq, are indeed sensitive to changes in the most abundant cell subsets (changes in gene expression and/or cell composition), but do not capture changes in rare cell populations. Recently, single-cell -omics technologies have been used successfully to study the immune cell responses triggered by BCG [57], HIV [83] and SARS-CoV2 [84] vaccines. Similarly, TCR and BCR sequencing have been applied at the single-cell level to characterize the immune repertoire of human individuals immunized with YF-17D [85], and influenza [86,87] vaccines and in HIV-vaccinated Rhesus macaques [88]. Similarly, Waickman et al. used single-cell RNA-Seq in combination with longitudinal TCR clonotype analysis to study T cell immunity in response to immunization with a recombinant, tetravalent dengue virus (DENV) vaccine [89]. Precisely, they were able to identify a set of biomarkers which characterize the most persistent vaccine-reactive memory CD8^+^ T cells. These studies illustrate how single-cell transcriptomic analyses can provide insights into the molecular mechanisms implicated in the regulation of immune memory and more generally, in immune responses to vaccines.

However, there are many challenges associated to the use of -omics technologies. The sample size for both discovery and validation cohorts needs to be sufficient to overcome the risk of high type 1 and 2 errors due to the large number of markers that are measured using omics technologies and the low contributing effect size of individual markers. Moreover, the integration of multi-omics data is far from being straightforward (see Section 5).

In addition to these high-throughput technologies, conventionally used in systems vaccinology studies, preclinical models offer the possibility to enrich vaccine signatures with imaging data.

## 4. Imaging Technologies to Refine and Expand Vaccine Signatures

### 4.1. In Vivo Imaging of Vaccine Trafficking and the Immune Response

The monitoring of vaccine components and the assessment of immune-cell dynamics at injection sites and lymph nodes allows better understanding of the immune response to vaccines. A variety of in vivo imaging modalities, including optical imaging (fluorescence and bioluminescence), magnetic resonance imaging (MRI), and nuclear imaging (positron emission tomography (PET), single photon emission computed tomography (SPECT)) can be used to study such dynamics. Thus, in vivo non-invasive imaging techniques are widely used to visualize the location and distribution of molecules, antigens, and inflammatory immune cells. The advantages, limitations, and applications of these imaging modalities have been well reviewed in the literature [90,91,92].

#### 4.1.1. Whole Body Imaging of Vaccine Distribution and the Immune Response 

Fluorescence imaging allows the in vivo visualization of the dynamics of vaccines and their interaction with immune cells in real time, facilitating the understanding of vaccine-induced immune-response mechanisms in preclinical models. For example, Romain et al. assessed the migration kinetics of a vaccine based on DCs expressing HIV-Gag protein from the injection site to draining lymph nodes (DLNs) in macaques by NIR (near infrared) fluorescence imaging [93]. Furthermore, Salabert et al. studied the development of in vivo and in vivo approaches to track vaccine-targeted Langerhans cells (LCs) following intradermal injection in NHPs [42].

Magnetic resonance imaging (MRI), widely used in clinical practice, makes it possible to obtain high whole-body anatomical resolution and is particularly suited for the analysis of both vaccine biodistribution and the associated immune response. This is achieved using various contrast agents and probes to label the vaccines and cells for MRI tracking [94]. For example, the in vivo longitudinal biodistribution of a vaccine labeled with superparamagnetic iron oxide (SPIO) was assessed in a mouse HPV16 tumor model and showed the presence of the antigen for several weeks post-vaccination in the DLNs [95]. The efficacy of DC-based vaccines is limited in patients and may be due to insufficient delivery of the vaccine to the lymph nodes. The tracking of iron nanoparticle-labelled DCs to the lymph nodes by MRI is possible in mice [96] and can be safely performed in patients [43], allowing the improvement of vaccine design.

The ability to directly image myeloid and lymphoid cells, and changes in their distribution in vivo is crucial to achieving a better understanding of the processes of the immune response. Several studies have demonstrated the ability to track and visualize immune cells by in vivo imaging (e.g., MRI and PET imaging) in various applications following either the reinjection of previously ex vivo labeled cells or the direct in vivo administration of specific labeled ligands [92].

The labeling of macrophages for the evaluation of inflammatory processes by MRI has been performed directly in situ after the phagocytosis of injected iron oxide contrast agents. This method has been shown to allow the visualization of tumor-associated macrophages (TAMs) or monocyte infiltration in various animal models [97,98,99,100]. Tremblay et al. [101] evaluated whether MRI can be used to track immune-cell populations in response to a lipid-based vaccine immunotherapy in a mouse model of human papillomavirus-based cervical cancer. They were able to track the increased recruitment of SPIO labeled CD8^+^ cytotoxic T cells and the decreased recruitment of myeloid-derived suppressor cells and regulatory T cells to the tumor with hypo-intensities due to the clearing of iron-labeled cells. However, the sensitivity of MRI is relatively low, limiting the possible detection of a low number of cells.

Nuclear imaging, PET, and SPECT are highly sensitive imaging modalities used in the clinic, based on the biodistribution of radiotracers within the body. SPECT imaging was proposed to investigate the biodistribution and kinetics of reinjected [^111^In] adiolabeled NK cell-based vaccines in patients with renal carcinoma. The authors observed the accumulation of 50% of the activity in the lesions, but a high level of circulating activity was also observed, caused by the released Indium-111 [102].

PET is mainly used in oncology for the visualization of sites of inflammation [90]. To date, only a few studies describing the use of PET for the tracking of vaccines have been published. Among them, Yuki et al. developed a PET imaging approach associated with MRI or CT (computed tomography) to study the biodistribution of an intranasal botulism antigen vaccine (Bo-Hc/A) labelled with [^18^F] in mice and NHPs [103]. In addition, Lindsay et al. developed an innovative dual radionuclide-near-infrared probe that allowed the longitudinal monitoring of an mRNA vaccine at the injection site and the lymph nodes, as well as its uptake by the immune cells, by PET and NIR fluorescence in macaques [41].

The visualization of inflammatory processes by PET imaging is an approach applicable to the monitoring of vaccine responses. Several studies have shown cell activation in lymphoid tissues, such as lymph nodes, by [^18^F]-FDG PET imaging after the administration of vaccines in mice [104] and humans [105]. Darrah et al. [57] described the use of [^18^F]-FDG, 2-deoxy-2-[fluorine-18]fluoro-D-glucose to track granuloma formation after Mycobacterium tuberculosis infection, as a correlate of active disease after BCG immunization. [^18^F]-FDG PET imaging can also be used for the monitoring of the inflammation related to vaccination, such as that induced by influenza vaccines [106,107,108].

Aarntzen et al. [109] showed the interest of using a radiolabeled thymidine analog, [^18^F]-labeled 3′-fluoro-3′-deoxy-thymidine ([^18^F]-FLT), to assess the proliferative immune cell response in lymph nodes after vaccination with an antigen-loaded DC vaccine. The authors showed a better correlation of [^18^F]-FLT than [18F]-FDG PET imaging with immune reactivity. However, theses imaging methods, lack the specificity and discriminatory power that antibodies or their fragments provide to specifically target immune cells in vivo [14,110,111,112].

Several studies aiming to track immune cells have used ex vivo *v* cell labeling to visualize them by nuclear imaging [113,114]. PET-CT imaging allowed the visualization of adoptively-infused NK cells, previously labeled with [^89^Zr]-oxine, in rhesus macaques. 

Ex vivo cell labeling shows certain drawbacks, such as the need of autologous transfer, especially in the case of clinical applications, or the potential loss of cell properties by ex vivo manipulation. Thus, other strategies have been used to directly label cells in vivo. Certain strategies have been recently developed to specifically target, track, and visualize disease-specific antigens, as well as immune-cell subsets, after injection of the antibody or derived fragments coupled with metal chelators, such as [^64^Cu], [^68^Ga], or [^89^Zr] [115,116] for PET (so called immuno-PET) or coupled with MRI contrast agents [117] or fluorophores for in vivo optical imaging [44,118].

Full-sized antibodies have been widely and successfully used for immuno-PET imaging [119,120,121]. However, the size and the long half-life of intact antibodies can be a limitation for their use as imaging agents. Many of these issues have been addressed by the use of smaller antibody fragments (Fabs, diabodies, single-domain antibody fragments (nanobodies), etc.) [112,116,122].

Among the strategies for imaging innate myeloid inflammatory cells, entire anti-CD11b, anti-class II major histocompatibility complex (MHC), and anti-macrophage mannose receptor antibodies or antibody fragments have been widely used to characterize inflammation by immuno-PET, mainly in mice [14,111,119,123,124]. For example, Cao et al. [119] developed the radiotracer [^64^Cu]-labeled anti-CD11b for longitudinal monitoring of the mobilization of CD11b^+^ myeloid cells from the bone marrow to the spleen and to local inflammatory lesions in mice.

Imaging of macrophages has already been performed in various applications to study inflammatory processes by targeting folate receptors [125] with radioligands. The macrophage mannose receptor has largely been used to track macrophages, especially with nanobodies specifically developed for SPECT and PET imaging to target the receptor in various preclinical models [123,126,127].

The presence of CD8^+^ T cells has also been monitored by immunoPET in preclinical tumor models, specifically in the context of immunotherapies using checkpoint-blockade inhibitors against the PD-1/PD-L1 and CTLA-4 axes [111]. Strategies can vary according to the injected radiolabeled antibody fragment [14,128,129,130,131]. An even higher specificity can be achieved by targeting and visualizing antigen-specific T cells in vivo [132].

Thus, whole-body immunoPET combines the sensitivity of PET with the high specificity and affinity of monoclonal antibodies. Furthermore, the use of antibody-derived fragments allows better tissue penetration, a lower background, and a smaller radiation burden for the patient.

#### 4.1.2. In Vivo Microscopic Imaging of the Interactions between Vaccines and Immune Cells

The complexity of the immune system, particularly when vaccines are involved, requires real-time, high-resolution imaging to visualize immune-cell interactions at the microscopic level. Intravital microscopy (fibered confocal fluorescence microscopy (FCFM), two-photon imaging) provides the detailed visualization of vaccines and their behaviors in the injection sites or lymph nodes.

Fibered confocal fluorescence microscopy (FCFM) was limited to preclinical applications due to the lack of human validated fluorescent tracers. FCFM is developed notably for the visualization of tumor growth and angiogenesis [133,134], as well as the tracking of vaccines and immune cell behavior. For example, Mahe et al. tracked percutaneous injected MVA expressing green-fluorescent protein (eGFP) in mice, its uptake by antigen presenting cells (APCs), and their transport to lymph nodes using FCFM [135]. Later, Rosenbaum et al. evaluated the kinetics of the arrival of MVA-eGFP-expressing cells in the skin by repeated in vivo imaging using FCFM (CellVizio Dualband®, Mauna Kea Technologies, France) in NHPs [36]. FCFM has also been used to study the in vivo effect of electroporation on antigen expression and APC behavior after intradermal injection of DNA-SIV vaccine expressing eGFP in macaques [118].

Two-photon microscopy (2PM) has been adopted to study single-cell dynamics at high spatial resolution at a depth of several hundred microns [136]. The available adapted 2PM devices allow ex vivo studies of tissue and in vivo studies in small animals. After vaccination, Rattanapack et al. studied the kinetics of in vivo uptake of a peptide antigen delivered within a lipid nanosystem by dermal DCs over time in mice [137]. It is also possible to track T- and B-cell motility in isolated lymph nodes from mice following antigen stimulation [138]. Bousso et al. used 2PM to directly examine the cellular dynamics of fluorescently labeled CD8^+^ T cells and DCs in vivo in the lymph node before and during antigen recognition in mice [139]. Although this approach is only limited to preclinical studies, intravital microscopy of the skin or surgically exposed internal organs offers excellent resolution for studying individual cells or even subcellular structures and microorganisms [140,141].

### 4.2. Ex Vivo Multiparametric Analyses

To provide the most advanced technologies for characterization of the complexity of the immune response to vaccination in relevant tissues, it is necessary to complement and strengthen in vivo imaging methods with state-of-the-art technologies for in situ multiplexed characterization of cellular and subcellular markers of the immune response. Despite the considerable advantages of available imaging techniques, one of the main drawbacks is the small number of parameters that can be simultaneously analyzed. 

Immunohistochemistry (IHC) and immunohistofluorescence (IHF) are currently the most common and suitable techniques to gain insights on changes in the spatial immune phenotype in various tissues. Immunofluorescence is widely used to detect multiple immune and inflammatory cell populations, as well as pathogens or vaccines in ex-vivo samples. It allows the assessment of innate and adaptive immune responses, especially their effectors in lymphoid organs or infected sites after pathogen challenge or immunization. For example, Darrah and al. [57] observed differences in immune-cell activation in the lungs of rhesus macaques depending on the route of BCG immunization.

Epifluorescence and confocal microscopy allow the observation of cellular and subcellular compartments. However, the number of observable markers is limited using such devices due to fluorophore spectra overlap. This can be partially resolved by the use of white laser confocal microscopy, spectral imaging, or cyclic immunofluorescence [142].

Tissue clearing, by reducing light-scattering and light-absorbing components, overcomes the limits of light penetration and thus allows deep imaging [143]. Three-dimensional imaging can then be performed by confocal, super-resolution confocal, multiphoton, and light-sheet microscopy [144]. For example, Li et al. [145] showed the interest of this method in characterizing the distribution of fluorescent vaccine constructs and the structural composition of tissues using an endothelial marker (CD31), T (CD8) and B (B220) lymphocyte markers. Although multiplex analyses can be performed at high spatial resolution using IHF approaches, the revelation of the diversity of the visualized markers can be limited.

Mass spectrometry imaging (MSI), the combination of molecular mass analysis and spatial information, can provide information on the spatial distribution of endogenous and exogenous species in tissue sections, without the need to disrupt sample integrity [146,147]. It enables both untargeted and direct targeted investigations for the discovery of disease or host immune response-related biomarkers, although the sensitivity for intact macromolecules, such as proteins, is still limited [148,149]. Matrix-assisted laser desorption/ionization (MALDI) is the most popular ionization technique for MSI due to its ability to image a wide range of molecular weights and molecular species (e.g., metabolites and proteins). To further analyze tissue specimens, multimodal studies combining MRI, 3D-MALDI-MSI, and histology [150] allow rapid correlation between molecular information and anatomical annotation. Combining MSI with liquid chromatography coupled to tandem mass spectrometry (LC-MS/MS) has also allowed the multiparametric analysis of proteins, lipids, metabolites, and mRNA to explore early immune events following vaccination [53]. The combination of MSI and histology is of particular interest for providing a snapshot of the tissue microenvironment and enables the correlation of drugs, metabolites, lipids, peptides, and proteins with histological/pathological features [151,152].

One of the major recent advances of MSI is associated with the introduction of mass cytometry imaging (MCI), a label-based MSI technique to follow protein markers at the cellular and subcellular level that combines mass cytometry and immunocytochemistry (ICC), IHC or IHF techniques, a high-resolution laser ablation system, and a low-dispersion laser ablation chamber [153,154]. MCI is a multiplex method for tissue phenotyping, signaling pathway imaging, and cell marker assessment at sub-cellular resolution (1 μm) that allows up to 40 parameters to be visualized in a single tissue section [155]. They offer the opportunity to investigate the heterogeneity of tissues and to understand disease development using disease-related probes [153,156,157]. MCI is currently more compatible with common sample preservation methods (formalin fixation or embedding in optimal cutting temperature compound (OCT)) [158]. It is being extensively used for the analysis of immune-cell composition, interactions, and localization in tissues. For example, this method allowed mapping of the anatomical location of myeloid cell subsets in human tonsil tissue [159] and the spatiotemporal relationship between memory B cells and the marginal zone [156].

Progress in the development of these high-multiplex techniques has allowed the identification of various cell populations that comprise the immune system, but they still present certain limitations. Gerner et al. developed an approach, histocytometry, which associates the spatial information that can be obtained using histology with the phenotypic data provided by flow cytometry [160]. They showed that the positioning of DC subsets within lymph nodes defines different levels of T-cell activation in response to vaccination [161]. Histocytometric analysis of human lymph nodes during HIV infection led to the identification of potent CD8 T cells within the germinal center, which could be considered as an effective component for the development of HIV cures [162]. Nevertheless, the non-uniform distribution of cells in the organ and the low density of certain populations compromise this analysis, performed on individual tissue sections. To overcome these limitations, Li et al. proposed methods to enable quantitative visualization of cells in their microenvironment within large tissue volumes, allowing better exploration of cellular relationships in various tissues [145,163]. Recent automation has allowed high-throughput image analysis, making histocytometry more useful for immunology applications [164].

High multiplexing methods have prompted the need for the development of image processing and analysis tools using complex machine-learning algorithms.

## 5. Bioinformatics and Statistical Tools to Build Predictive Models of Vaccine Responses

### 5.1. Analysis of High-Dimensional Biological Data

The ability to efficiently use or design analytical pipelines for interpreting -omics data has emerged as a critical element for modern vaccinology research. Such analytical pipelines are generally developed using one or more programming languages (usually including R or Python). Their aim is to handle a large panel of analytical steps, ranging from data preprocessing to data integration. Proper analytical pipelines should be sufficiently flexible to tackle various studies but also sufficiently focused on certain methods to answer immunological questions specific to each study. Bioinformatics pipelines used to interpret high-dimensional immunological data should follow the FAIR principles (that is to say that data collected and handled within these pipelines must be findable, accessible, interoperable, and reusable) [165]. The use of paradigms derived from computer science, such as code versioning and the writing of extended documentation, are fundamental when developing pipelines, as it allows them to be reusable over time. Machine-learning methods and algorithms that comprise analysis pipelines can generally be classified into unsupervised or supervised approaches. Unsupervised algorithms aim to analyze datasets without major a priori assumptions, especially in terms of the biological conditions to which the samples belong. The most commonly used unsupervised algorithms are clustering (kmeans, hierarchical clustering), dimensionality reduction (principal component analysis (PCA), multidimensional scaling (MDS)), and association rules learning (a priori algorithm). On the other hand, supervised algorithms aim to analyze datasets with direct consideration of the metainformation available for each sample. The most commonly used supervised algorithms are discriminant analyses, decision trees with random forests, and, more generally, all classification or regression approaches. Although supervised analyses are critical for identifying biomarkers for an immunization, unsupervised analyses are critical for revealing unexpected characteristics of a dataset (such as subgroups of responders/non-responders and the heterogeneity of the conditions). Both unsupervised and supervised approaches are extremely complementary when analyzing -omics responses to vaccines and immunizations.

Due to its development in the early 21st century, transcriptomics analysis now benefits from an outstanding variety of algorithms, methods, software, and dedicated databases. Despite the large set of available data analysis approaches, differential expression analysis is still the gold-standard for interpreting transcriptomic profiles [166]. Differential expression analysis aims to identify genes or transcripts that are significantly differentially expressed between conditions to identify biomarkers of the immune response. Volcano-plot representations are informative graphs used to visualize the magnitude (quantified using the relative fold-change of expression) and statistical significance (quantified using *p*-values) of differentially expressed genes. Once identified, the ability of one or multiple gene signatures to segregate the biological conditions of interest is tested using multivariate representations. Among them, heatmaps combined with dendrograms show the relative levels of gene expression for the gene signature in all samples using unsupervised hierarchical clustering at both the gene and sample level. Dimensionality reduction methods, such as PCA and MDS, are also useful in determining the quality of a signature and its ability to separate conditions in a multivariate manner. Venn diagrams are common representations that show the amount of overlap between multiple gene signatures. Due to their length (generally ranging from a few hundred to a few thousand genes), gene signatures cannot be interpreted manually, gene by gene, in relation to the literature and must be interpreted using specific methods. Functional enrichment analysis gathers a large set of methods and databases and aims to identify over-represented biological functions or pathways in a gene signature of interest. Statistical tests, generally based on Fisher’s exact test, make it possible to determine which pathways are significantly over-represented. The most widely used databases for functional enrichment analysis are Gene Onthology [167], KEGG, and WikiPathways. Other databases, such the Human Gene Atlas database [168], are of interest in systems immunology as they aim to deconvolute transcriptomic profiles from PBMCs or whole blood samples into the cell populations that comprise them. The EnrichR database [169] is a meta-database for functional enrichment analysis that is composed of more than 170 different databases of pathways, function, and biological properties. Such a large spectrum of covered databases is useful for the discovery of functions associated with identified gene signatures. Gene co-expression networks, created by algorithms such as WGCNA [170], can be used to identify sets of genes with similar expression patterns in the datasets to complement the analysis of transcriptomic profiles. In such networks, each dot corresponds to a gene and genes are linked if there is a significant correlation between their expression profiles in the dataset. Such approaches are especially useful when integrating transcriptomics data with other -omics or clinical information.

The development of algorithms for analyzing cytometry profiles is still an active area of research. Although automatic gating approaches mimic cytometry experts by positioning gates in cytogram plots (two-dimension (2D) representations in which each axis corresponds to the expression of one cell parameter), automatic cell clustering algorithms identify groups of cells that have similar phenotypes (also called cell clusters). SPADE [171] and FlowSOM [172] were among the first widely adopted automatic cell clustering algorithms. Dimensionality reduction methods combined with unsupervised clustering are now commonly used, especially the tSNE [173] and UMAP [174] algorithms, which generate 2D representations of cytometry profiles and have become increasingly popular in recent years in immunology and vaccinology. In such 2D representations, each dot corresponds to a cell and cells are positioned based on their similarity of expression for selected markers. Once generated, UMAP or tSNE representations can be overlaid with the expression of specific markers using a color gradient to annotate cells and define sets of cell subpopulations. Clustering algorithms can be used to automatically identify these groups of cells. Such algorithms are essential, as they can identify complex phenotypes of cell populations that cannot be characterized using regular manual gating approaches. Importantly, these algorithms can also identify cell populations that have distinct phenotypes, as well as those that show continuous differences in marker expression (especially important in the context of cell differentiation and activation states). Once cellular clusters have been determined, the aim of subsequent analyses is to identify clusters that are statistically differentially abundant between conditions. Topological data analysis algorithms are currently used to unravel the characteristics of cell differentiation or kinetics to a stimulus. The annotation of determined cell clusters (also called cell cluster labeling) is currently a major challenge. The aim of such approaches is to annotate the cell clusters based on their levels of marker expression and existing knowledge about the cell populations. The exact classification of cell populations into a well-defined nomenclature does not yet exist and represents a major limitation for applying these annotation algorithms. In addition, the complexity and heterogeneity of the cell populations involved in vaccination are yet to be fully explored.

Single cell sequencing will have for sure a pivotal role in modern biology to decipher both molecular and cellular events involved in vaccination. Thanks to this technique, key internal mechanisms responsible for cell activation, proliferation, and differentiation will be characterized at unprecedented level of detail allowing more rational when designing vaccines. While most of the recent efforts have been done for applying this technique on transcriptomics, the characterization of B and T cell repertoires at the single cell levels is of great interest. The analysis challenge for single cell sequencing data is important as methods created for the analysis of bulk transcriptomics and high dimensional cytometry must be combined for handling them. The Cell Ranger suite developed by 10X Genomics allow bioinformaticians a straightforward way to analyze single cell sequencing data, especially regarding the preprocessing steps. The first analysis step consists of the alignment of sequenced reads of a reference genome. The filtering of cell events with abnormal number of mapped transcripts or associated with aberrant mitochondrial activities is done at this step. Once the reads have been aligned on the reference and transcript expressions are quantified for each cell of each sample, an analysis step consisting in a dimensionality reduction is done. As for cytometry data, tSNE or UMAP algorithm are commonly used for that purpose. Of note, the Loupe browser also developed by 10X Genomics offers the possibility to graphically handle UMAP and tSNE representation for a processed dataset. Different R packages or approaches, such as Seurat [175] are complementary to cell ranger and Loupe browser for interpreting these data. Efforts are now done to create methods and algorithms able to integrate events of different structure together, allowing then a holistic characterization at a single cell resolution.

### 5.2. Machine Learning and In Silico Models

Machine learning is “the study of computer algorithms that improve automatically through experience”, as defined by Tom Mitchell [176], by learning from the data to make predictions about the data. Machine learning is widely used in various applications in biology. It allows the solving of complex problems using observations or data. Machine-learning algorithms can be classified into three types: supervised learning, unsupervised learning, and reinforcement learning. Various supervised machine-learning algorithms can be used to predict the immunogenicity, efficacy, or reactogenicity of vaccines, either by performing classification, which is a predictive modeling approach in which the output of data is composed of class labels (discrete values), or by performing regression, which is also a predictive modeling approach, but the output is in the form of quantities (continuous values). There are a large number of algorithms available that can be used to predict biomarkers of vaccine responses. However, choosing one over another can be challenging, as the choice depends on various considerations, such as the amount of data, its interpretability and accuracy, training time, number and type of features, and many other factors. Thus, to conduct a scientific study using machine-learning algorithms, one must prioritize the considerations that are the most relevant to the study and questions addressed and proceed by implementing the most pertinent algorithms and comparing them. Importantly, before using any machine-learning algorithm for prediction, a process of data cleaning and processing and feature selection is required. This is a key step but is not the focus of this review. However, this topic has been recently reviewed elsewhere [177]. Table 1 summarizes the principles of several machine learning methods and their respective pros and cons (Table 1).

Several vaccines have been studied with the goal to identify predictors of immunogenicity after delineating signatures that correlate with immunogenicity, mainly in healthy adults (Table 2). In most cases, it consists of predicting the magnitude of the antibody response, which is often a correlate of protection, with early predictors induced within the first week following immunization. However, certain studies have aimed to find pre-existing predictors before immunization [60,62,178,179], or predictors of the intensity of specific T-cell responses [52,180], protection after experimental human challenge infection [181], or reactogenicity [182,183]. Most studies identified predictive genes or gene sets (from PBMC/whole microarray or RNA-Seq). However, more recent studies have used additional molecular data, such as metabolite clusters and cytokines, as well as cell populations, in addition to gene transcripts to predict the antibody and T-cell responses to the live-attenuated VZV vaccine for example [180]. Variables appeared to be highly connected or even overlapping in the so-called multiscale, multifactorial response network (MMRN) that was constructed to integrate the multi-omics data. The authors proposed that the MMRN approach increases the statistical prediction beyond linear models by network connections that accommodate indirect steps and temporal developments.

The most popular vaccine models for immunologists include the YF-17D and flu vaccines. YF-17D represents an ideal vaccine to understand and mimic because it induces life-long protection after a single injection. There are two major types of influenza vaccine: a live attenuated vaccine, which is delivered intranasally, and an inactivated vaccine, which is injected intramuscularly, both providing protection through distinct mechanisms. Adjuvanted versions or higher doses are also available for specific populations, such as the elderly, and thus represent the first personalized vaccines. However, these traditional seasonal flu vaccines do not provide long lasting and broad protection. They are based on yearly predictions of the circulating viruses, and they confer protection only when strains do match the circulating viruses. A key challenge is to develop pan-influenza viruses vaccines targeting conserved regions that would protect against seasonal, future drifted and pandemic strains. Anyway, not surprisingly, YF-17D and seasonal trivalent inactivated influenza vaccines (TIVs), likely because of practical reasons (annual immunizations of adults with a safe, albeit imperfect vaccine and identified immune correlate of protection), are over-represented among studies to define predictors of immunogenicity. Of note, predictors of the antibody and T-cell responses [52,180] and baseline and early predictors [60,185] of a given vaccine and study differ. Among the genes present in the various DAMIP gene signatures predictive of the antibody response to vaccination with TIV and YF-17D, seven are shared [52,185]. Finally, the predictive signatures have also differed for TIVs, depending on the season, the age and health status of those vaccinated, and the machine-learning method.

It is expected that predictors of vaccine-induced antibodies could be clinically useful by predicting suboptimal immune responses to certain vaccines to stratify them, for example those requiring a booster. However, the robustness and predictive accuracy depend on the sample size and the identification of solid predictors requires extensive validation in multiple clinical trials. A cost-effective PCR-based ‘vaccine chip’ that focused on a set of predictive genes was successfully developed for flu vaccines [185]. It is admittedly more challenging [195], but predictors can also provide insight about key players (molecules or cells) and uncover new mechanisms to target to more rationally improve vaccines. Several studies to identify predictors of vaccine immunogenicity included or were followed by mouse studies to evaluate the mechanistic relevance of the predictors, including *Camk4* [185] and apoptosis [178] for flu vaccines and *Gcn2* (also known as Eif2ka4) for YF-17D [196].

The purpose of mechanistic mathematical modeling differs from that of machine learning. It aims to mimic biological mechanisms through observations of and assumptions about the phenomenon of interest. This type of modeling uses mathematical formulations that seek to identify a mechanistic relationship between inputs and outputs of the phenomenon of interest [197,198]. These approaches are complementary to machine-learning approaches, which seek to establish statistical relationships and correlations between inputs and outputs. Due to the oversimplified assumptions and extremely specific nature of mechanistic mathematical models, they are limited to establishing universal predictions, which are achievable by machine learning. However, mechanistic modelling may be more suitable for studying certain phenomena than machine-learning approaches, depending on the research objectives. Therefore, these two approaches should not be considered as competing with each other but rather as complementary [199].

## 6. Vaccine Signatures in Preclinical Models to Improve Human Vaccination Strategies

Although reducing and refining animal experiments require permanent efforts from the scientific community, assessing vaccine responses in animal models is still, for now, a necessary step in the vaccine registration process [200]. Currently, preclinical trials often provide key decisional points to pursue vaccine development [201], as it has been the case for SARS-CoV-2 vaccine candidates [202]. Indeed, they allow to design robust, controlled studies with a wider range of tools and samplings than the ones available in clinical trials. Plus, some models, such as NHPs, are highly predictive of the human immune and vaccine responses.

Applying systems vaccinology approaches to animal models assuredly provides a way to improve preclinical studies, accelerate vaccine development, and increase our knowledge of vaccine-induced protective responses, as discussed below.

### 6.1. Defining New Correlates of Protection

Through the use of diverse high-throughput technologies, systems immunology can lead to the identification of multiple biomarkers, combined into complex signatures that can be used to build predictive models of vaccine-induced responses. Being able to predict the efficacy of a vaccine with the identification of early biomarkers could considerably facilitate preclinical and clinical trials. Currently, vaccine development starts with an exploratory phase that addresses basic scientific questions and ideally leads to proof-of-concept studies that validate the efficacy of the vaccine in experimental—usually small animal—models. Translational research then follows with immunogenicity and efficacy studies in preclinical models, which guide subsequent clinical trials in humans [203].

The evaluation of vaccine efficacy can be challenging and partially relies on the possibility to define reliable correlates of protection. According to Plotkin and Gilbert, a correlate of protection can be defined as a marker of immune function that statistically correlates with protection after vaccination [204]. As mentioned previously, antibody titers have been used as correlates of protection for many vaccines. For example, a protective level of antibody measured by ELISA has been defined for vaccines against hepatitis A virus (HAV) and HBV [205]. For other vaccines, such as rabies or YF vaccines, neutralization titers have been linked to protection [206,207]. While antibodies can be considered as reliable correlates of protection for genetically stable pathogens, they cannot be properly used in the case of viruses with high mutational capacities and that can escape humoral responses, as we observe in the still on-going SARS-CoV-2 pandemic [208]. In addition, well-defined correlates of protection are still lacking for multiple vaccines, including those against malaria and tuberculosis [209], and serological measurements are probably not the only or relevant immune correlates.

Defining accurate correlates of protection is pivotal in the vaccine development process, as it allows the rapid assessment of vaccine efficacy. Hence, applying systems immunology approaches to clinical trials has been proposed to accelerate the discovery of early predictive markers of vaccine efficacy [210]. Additionally, systems vaccinology could be integrated into preclinical studies and further empower translational vaccine research. Cellular immunity will need to be explored thoroughly to enable better prediction of vaccine immunogenicity. As already discussed, local cellular events shape subsequent protective responses [36,211,212,213]. Thus, their characterization can provide a way to rapidly assess the quality of vaccine-induced immunity. In-depth studies of immune cells involved at the site of vaccination, such as skin DCs or LCs, is achievable in preclinical models due to the use of multiple and complementary exploratory approaches (Figure 2) and can provide very early predictive markers of vaccine immunogenicity. Elsewhere, in addition to antibody responses, exploring T cell responses will certainly provide key biomarkers of efficacy for vaccines targeting mutating pathogens. For example, T lymphocytes seem to play a key role after SARS-CoV-2 infection and vaccination [214,215]. Precisely, a study on Rhesus macaques showed a strong decrease of the protection induced by natural immunization after depletion of CD8+ T cells thus indicating that CD8+ T cell responses could be used to define more accurate correlates of protection for SARS-CoV-2 infection [216]. Additionally, preclinical models are often instrumental in appraising long-term immune memory, as longitudinal studies can be initiated and conducted more easily and rapidly than in human populations. For example, NHPs have been successfully used to study memory responses induced by vaccines against SIV, tuberculosis, and or polio [59,217,218,219]. Implementing systems vaccinology techniques in prolonged longitudinal preclinical studies [72] could link initial and long-term responses and generate early signatures of immune memory. 

Applying these approaches to human studies could therefore provide a way to reduce the time and cost of vaccine efficacy trials.

### 6.2. Stepping up Personalized Vaccinology

Presently, validated correlates of protection are generally based on results obtained from cohorts of vaccinated healthy adults, whereas target human populations are often highly heterogeneous, and defining a universal response to vaccination is probably impossible [220]. Indeed, many host-related factors can modulate immune responses to vaccination, including age, gender, infectious history, comorbidities, genetic background, and microbiota composition (Figure 2) [28].

In vulnerable populations, such as infants and young children, pregnant women, and elderly and immunocompromised individuals, immune responses to vaccination are often under characterized and assessing vaccine efficacy in these specific populations remains a major challenge. Preclinical models are well-suited for in-depth studies on population subsets with specific attributes, especially animal models with the most predictive value for human vaccine efficacy studies, such as NHPs. Although systems vaccinology approaches have already been used in human cohorts of aged individuals [63,221,222], high-throughput characterization of vaccine-induced responses in children are still rare [9,223] and awaited for newborns [224]. Importantly, NHPs provide a highly relevant pediatric model to test vaccine efficacy, as they share many similarities with humans in terms of immune system development [225,226,227,228].

Infant NHPs have been frequently used in studies to evaluate vaccines against Mycobacterium tuberculosis [229,230,231] and HIV [232,233,234]. Recently, Han et al. used high-throughput technologies, including single-cell RNA sequencing of PBMCs, to perform an in-depth comparison of neonatal and adult immune responses to HIV immunization in macaques [235]. More specifically, they were able to show higher activated circulating TFH cell frequencies in Env-immunized neonatal macaques than in adults. This study also revealed distinct post-immunization transcriptome profiles between infant and adult macaques, with elevated B-cell lymphoma 2 (*BCL2*) transcript levels in T cells and lower interleukin-10 (*IL-10*) receptor alpha (*IL10RA*) transcript levels in T, B, and NK cells and monocytes of macaque neonates. This study illustrates how systems vaccinology in preclinical models can guide and help human vaccine efficacy trials. Moreover, identifying vaccine signatures in preclinical models of vulnerable populations would allow investigation of the influence of host factors on responses to immunization.

Pharmacogenomics and vaccinomics also contribute to improve personalized medicine and vaccination, respectively [19,20]. Indeed, identifying genetic polymorphisms associated to increased or decreased vaccine efficacy before administration of the vaccine may allow to adapt the vaccination strategy to particular individuals or to sub-groups of different ethnic ancestry. However, human GWASs require large cohorts of thousands of individuals to detect true genotype-phenotype associations [236] and are thus limited to the retrospective study of licensed vaccines. Vaccinomic studies in preclinical models could provide a way to rapidly assess the influence of host genetic factors on the efficacy of newly developed vaccines. For ethical and financial reasons, vaccine preclinical studies in NHPs commonly use limited numbers of individuals and are thus not powered for genetic association studies. However, alternative models such as genetically diverse mouse populations are well-suited for this application. For example, the Collaborative Cross constitutes a new experimental platform to investigate the influence of host genetic factors in the susceptibility to infectious diseases [237] and starts to be used to study vaccine-induced responses [148].

Consequently, the various approaches of systems vaccinology empower personalized vaccination strategies, which should improve efficient vaccination coverage in target populations.

### 6.3. Improving Vaccine Formulation and Administration

Another way to increase global vaccine efficacy in highly diverse human populations is to improve vaccine formulation with adjuvants and refine administration routes and schedules. Systems immunology offers powerful tools to investigate the effects of adjuvants and vaccine regimens on human immune responses [9,181] and can lead to reference signatures to which new vaccine candidates can be compared. Additionally, preclinical models are still highly valuable for the development and evaluation of innovative adjuvants, new administration routes, and various vaccine regimens.

As previously stated, applying systems biology technologies in this context will further improve preclinical studies and enhance innovation and progress in the field of vaccinology. For example, transcriptomics was first used in mouse models to characterize the molecular and cellular signatures of clinically tested human vaccine adjuvants [26,238]. These studies provided new insights about the modes of action of vaccine adjuvants by identifying both common transcriptional differences and adjuvant-specific responses associated with either germinal-center reactions or the orientation of helper T cell responses [26,238]. Systems vaccinology studies performed in NHP preclinical models can further guide the rational development of vaccine adjuvants for clinical use. Kasturi et al. studied the capacity of TLR agonists to promote durable protective immunity against SIV [239], whereas Thompson et al. investigated the effect of such TLR-based adjuvants on immune responses induced by a vaccine against *Plasmodium falciparum* [240]. In a huge leap forward, Chaudhury et al. combined extensive immuno-profiling of three adjuvant formulations in Rhesus macaques with multivariate analysis and machine learning [32]. This study identified adjuvant-specific immune “fingerprints” that could be used as rudiments to define correlates of protection and immunogenicity in human vaccine trials.

The mode of vaccine administration is another important parameter that modulates immune responses, as illustrated in a recent study of Cirelli et al. on the effect of slow delivery of HIV antigens in Rhesus macaques [88]. They showed that slow-delivery immunization improves neutralizing antibody production and leads to increased frequencies of antigen-specific GC-TFH cells using a combination of high dimensional techniques, such as scRNA sequencing, BCR sequencing, and whole lymph node imaging. In another study, Adam et al. investigated the early mechanisms that occur in the skin after intradermal injection and electroporation of SIV immunogens in Cynomolgus macaques. They used flow cytometry, cytokine profiling, and transcriptomics of skin cells to demonstrate that electroporation has a strong adjuvant effect mediated by inflammatory cell recruitment and LC mobilization [212]. Finally, studying how the vaccine regimen influences immune signatures could be critical to improving vaccination strategies in human populations. For example, mass cytometry was used to perform in-depth phenotyping of innate immune cell populations differentially induced by MVA vaccine prime and boost immunizations [34,35]. Similarly, multi-parameter flow or mass cytometry were successfully used to study the effects of heterologous prime-boost combinations of tuberculosis vaccination in mice [241], and the influence of the interval between MVA immunizations in Cynomolgus monkeys [33].

Overall, these results illustrate how systems vaccinology can strengthen preclinical findings on vaccine formulation and administration, and thus support vaccine development processes.

### 6.4. Deciphering Mechanisms That Underly Immune Protective Responses

Finally, the use of systems vaccinology in preclinical models will provide mechanistic insights on immune responses triggered by vaccines. Because of the constraints on collecting tissue samples in human clinical trials, most studies have strived to define signatures of myeloid and lymphoid responses in the blood [8,10,61,62,142,189]. However, immune processes are highly orchestrated in time and space and thus occur in multiple tissues and organs of the body (Figure 2). 

As mentioned previously, early local events in the skin following immunization can shape subsequent immune responses and vaccine efficacy [36,211,212,213]. In other respects, the acquisition of immune memory characterizes adaptive responses and is of critical importance for vaccines to confer long-lasting protection. Specifically, activated B and CD4^+^ T cells differentiate into memory cells within lymph nodes and spleen GCs, and then migrate to the bone marrow, which represents their main homing site [242,243]. Thus, a major challenge for human vaccine studies is to identify signatures in the blood that would truly reflect the magnitude and persistence of tissue-specific immune events. Recent efforts have been undertaken to investigate vaccine-induced immune events in human skin explants [53] or in human lymph node fine-needle aspirates [86,244]. However, preclinical models are still instrumental to comprehensively studying the mechanisms of vaccine-induced immune responses at the tissue and whole-body levels (Figure 2). For example, immunogens slow delivery administration, described by Cirelli et al., triggered higher frequencies of HIV-specific GC B cells and altered their BCR repertoire compared to conventional bolus immunization [88]. In another recent study, Eslamizar et al. examined NHP lymph node-derived cells after immunization with HIV Env protein encoded either by a plasmid DNA, a recombinant MVA, or a recombinant VSV vector [245]. They demonstrated that recombinant MVA was the most potent vector to induce GC-TFH cells expressing high levels of ICOS (inducible T cell co-stimulator), a key receptor of TFH help to GC B cells. The investigation of how mucosal immunity is modulated by immunization and contributes to vaccine-induced protection is another field of research in vaccinology that warrants the exploration of tissue-based immune processes. For example, systems immunology approaches have been used to characterize the molecular signatures of vaginal tissues of macaques immunized with a TLR-adjuvanted SIV vaccine [239] and to show that vaginal HIV Env-specific antibody and cellular responses presumably confer auxiliary mechanisms of protection against viral challenge in NHPs [83].

In addition, animal models are also particularly appropriate for assessing and dissecting immune responses to a combination of vaccines and/or pathogens. Indeed, exposition to pathogens and the vaccination history of animals are strictly monitored and recorded in laboratory settings. Such controlled experimental conditions facilitate the study of interactions between antigenically unrelated pathogens or immunogens. Recently, much attention has been given to the importance of non-specific effects of vaccines with a growing body of evidence suggesting that live-attenuated vaccines such as BGC, measles and oral polio vaccines overall improve childhood health [246]. Indeed, several randomized trials and population-based cohort studies led in young children revealed a significant decrease in infectious disease mortality rate in BGC-vaccinated newborns in low-income regions and a reduction of the risk of admissions for infectious diseases in BGC-vaccinated babies in high-income settings [246]. Several hypotheses have been formulated to explain the underlying mechanisms, including cross-protection conferred by heterologous T cell responses and the development of a kind of innate immune memory now referred to as “trained immunity” [247,248,249]. However, much remains to be done to fully understand the precise mechanisms underpinning non-specific effects of vaccines. To this end, preclinical models can greatly contribute to increasing our knowledge on trained immunity processes, as recently reviewed by Palgen et al. [250], and systems vaccinology will also endorse such studies.

Finally, although NHPs most likely provide the best model of human vaccine responses in terms of prediction, small animal models can be used to further investigate the functional mechanisms underlying predictive biomarkers identified in preclinical and clinical studies. Notably, transgenic mice have been instrumental in elucidating the functions of genes implicated in immune responses to pathogens or vaccines [251,252]. For example, Querec et al. identified general control non-derepressible 2 kinase (GCN2) as a biomarker that correlates with CD8^+^ T-cell responses to YF-17D vaccination in one of the earliest systems vaccinology studies [52]. They used mice carrying constitutional or conditional knock-out deletion of *Gcn2* to demonstrate that GCN2 leads to increased autophagy and antigen presentation in DCs in response to YF-17D immunization and thus revealed a connection between a vaccine-induced stress response in DCs and adaptive immune responses [253].

To conclude, preclinical models are invaluable for expanding our knowledge of immune mechanisms underlying vaccine protection and for improving rational vaccine development.

## 7. Conclusions

Systems immunology approaches used in preclinical studies and clinical trials are invaluable in vaccine biomarker discovery through the analysis and statistical modeling of large datasets.

Despite the outstanding development of high-throughput technologies and computational methodologies, many challenges still remain to be tackled to realize the full promise of systems vaccinology. Indeed, most recent studies in the field of vaccinology have restricted their findings to a description of vaccine biomarkers without further exploring the correlative, predictive, or explanatory aspects of these signatures. Furthermore, being able to discriminate between markers of immunogenicity and protection is not an easy task as illustrated by many clinical trials of HIV vaccines [254]. The application of systems immunology, in particular systems serology, to preclinical and clinical trials of HIV vaccines offers a means to specify new correlates of protection, though we are still at the beginning of this process [29].

In this review, we highlighted numerous technological and biological aspects useful to study vaccine responses and to the development of future vaccines. Nevertheless, limitations inherent to research techniques or to animal models question the methods currently employed in vaccine studies. First, the inherent differences in data formats generated by immunophenotyping platforms are a major impediment to the integration of large data sets. For instance, mRNA expression levels do not always represent intracellular protein expression levels and may not either reflect extracellular marker expression detected by cytometry [255]. Second, the measurement of diverse immunological parameters usually implicates multiple platforms and/or laboratories, standardization efforts are thus required to reduce technical and analytical variability and improve the robustness of predictive models [256,257]. Third, collaborations between vaccinologists and researchers with expertise in bioinformatics and computational and mathematical modeling also need to be strengthened to create predictive algorithms that are transposable to clinical applications.

Although human clinical trials of vaccine efficacy increasingly rely on systems immunology approaches, preclinical studies are lagging behind. Nonetheless, preclinical models harbor rare assets to explore vaccine-induced responses, including the possibility to investigate immune processes in tissues and the whole organism. The future of defining comprehensive vaccine signatures will likely rely on extended data analyses, including data obtained through imaging technologies, similarly to what has already been implemented in the field of cancerology [258,259]. In the near future, we believe that it will be possible to optimize and standardize several high-dimensional technologies so that they can be used conjointly on a regular basis in preclinical and clinical trials. For example, mass cytometry and imaging mass cytometry employ the same reagents, rely on the same detection method and generate datasets which can be analyzed with the same bioinformatic pipelines while providing complementary biological information on vaccine responses. Finally, applying high-throughput technologies to preclinical studies will expand our knowledge of the immune processes induced by vaccines in experimental models and hopefully improve the rate of translation of discoveries from animal studies to human trials [219].

Overall, application of systems biology concepts to preclinical and clinical studies promises great advancements in our understanding of vaccine-induced immune protective responses and provides unprecedented opportunities for the development of new vaccines.

## Figures and Tables

**Figure 1 vaccines-09-00579-f001:**
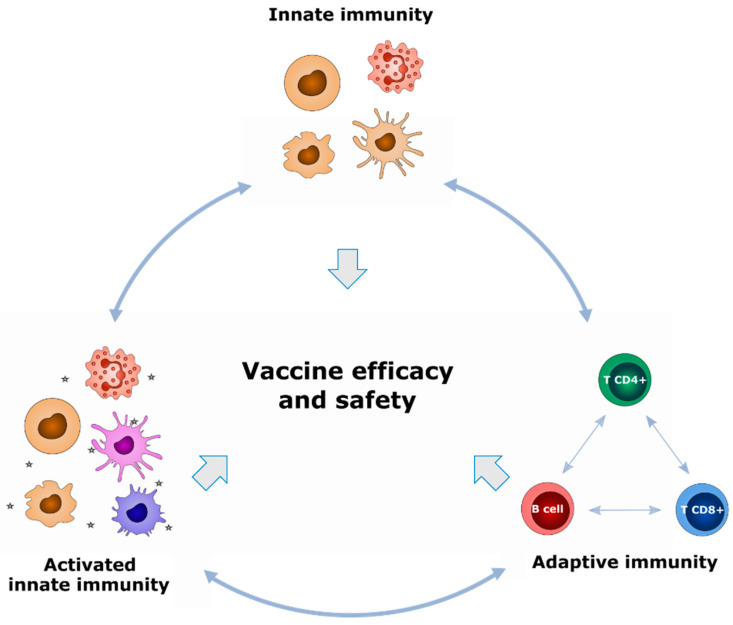
Vaccine efficacy and safety are determined by interactions between innate and adaptive immunity. These interactions are shaped by host factors and can be orientated by vaccine properties.

**Figure 2 vaccines-09-00579-f002:**
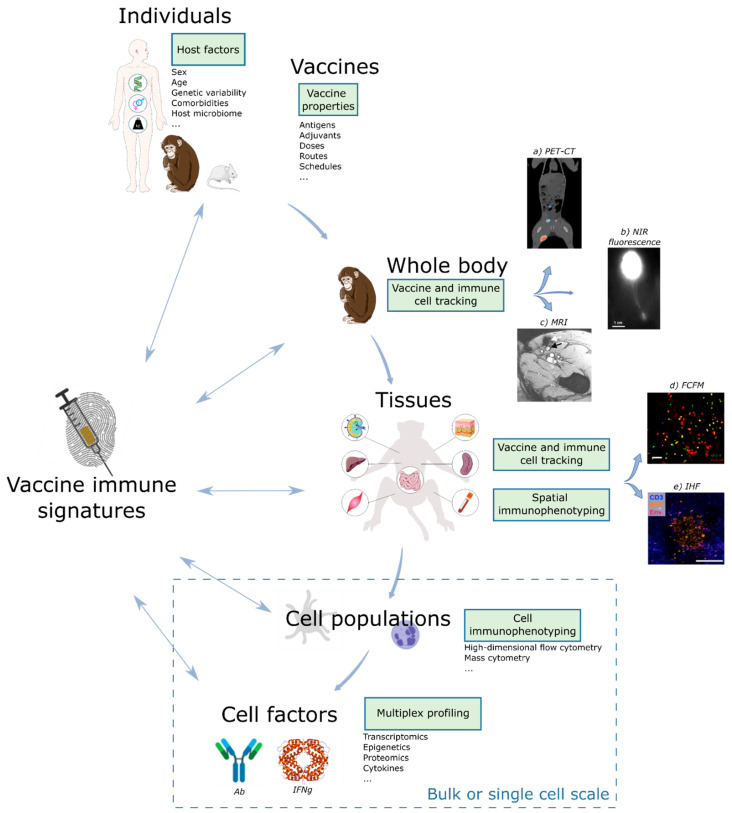
From individuals to single cells: integrating multi-level data into comprehensive vaccine signatures. Host factors and vaccine properties are important determinants of immune responses. Variations of these determinants, such as genetic polymorphisms, age, host microbiome or immunization procedure, thus condition the definition of vaccine signatures. Systems immunology enables the identification of biomarkers of vaccine responses at multiple scales, from whole-body to cellular factors. Diverse high-throughput technologies, including in vivo imaging, allow the characterization of vaccine immune signatures through various applications, such as immune-cell tracking, cell immunophenotyping, and multiplex profiling. Combining and integrating data at different scales will be of great value in identifying extensive vaccine immune signatures. (**a**) Positron emission tomography-computed tomography (PET-CT) imaging of the YF preM mRNA vaccine in NHPs [41]. (**b**) Near-infrared fluorescence (NIR) imaging to follow an anti-Langerin-HIVGag fusion vaccine from the injection site to the draining lymph node [42]. (**c**) Magnetic resonance imaging (MRI) of a DC-based vaccine in the lymph node [43]. (**d**) In vivo tracking of Langerhans cells within the skin by fibered confocal fluorescence microscopy (FCFM) [44]. (**e**) Tracking of fluorescently labeled HIV-1 envelope glycoprotein trimers in lymph nodes by immunohistofluorescence (IHF) [45].

**Table 1 vaccines-09-00579-t001:** Principle, advantages and drawbacks of common machine learning algorithms.

Machine Learning Algorithm	Principle	Advantages	Drawbacks
Linear regression	It assumes a linear relationship between input variables and output and thus, attempts to model this relationship by fitting a linear equation to the observed data There are several implementations of this model, of which the most commonly used is ordinary least squares, which tends to minimize the residual sum of the squares between the observed and predicted targets.	Simplicity Ease of implementation	It assumes that the input variables are independent It risks generating biased models due to oversimplification
Linear discriminant analysis (LDA)	It is used to identify to which class samples belong to, certain statistical properties of the data are first calculated and then substituted into the LDA equation. The statistical properties consist of the mean and variance for the case of a single input and the means and covariance matrix for multiple inputs.	Simplicity Robust and interpretable classification results	Does not perform well when the discriminant information is not present in the mean It cannot be applied to non-linear problems
Random Forest	It builds a number of decision trees on bootstrapped training sets and considers a random sample of m predictors to be split candidates from the full set of p predictors to overcome the problem of high variance. Therefore, on average, the strong predictor is not considered and other predictors have a better chance. This process can be thought of as decorrelating of the trees, thereby making the average of the resulting trees less variable and hence more accurate and reliable.	Reduced variation. Accurate and reliable It works well for both classification and regression problems	It requires considerable computational power and time for training It suffers from interpretability
Support vector machine	It converts a non-linear separable problem by transforming it onto another higher dimensional space and thus, the problem becomes linearly separable. This is accomplished using various types of so-called kernel functions. Then, classification is performed by finding the hyperplane that well separates the classes of samples.	It can solve any complex problem with the appropriate kernel function Less risk of overfitting	Choosing the appropriate kernel function is not easy It does not work well with large or noisy datasets
Discriminant analysis via mixed integer programming (DAMIP)	It is a classification model based on a very powerful supervised-learning approach used primarily in the biomedical field. It is a discrete support vector machine coupled with a powerful embedded feature-selection module [176].	It reduces noise and errors. It applies constraints that result in superior classification accuracy Universally consistent. Handles well imbalanced data	This algorithm is mainly used in the biomedical field, little is known about its drawbacks in literature

**Table 2 vaccines-09-00579-t002:** Machine learning methods to predict vaccine immunogenicity and efficacy. Different machine learning algorithms can be used. The quality of the model needs to be evaluated, and there are different metrics to assess a model performance, such as accuracy (defined as the number of correct predictions divided by the total number of input data), Area Under the Receiver Operator Characteristic curve (AUROC) or Root Mean Squared Error for regressions. It depends on the machine learning method itself. (Ab, antibody; ClaNC, classification to nearest centroid; DAMIP, discriminant analysis via mixed integer programming; HAI, hemagglutination-inhibition; CHMI, Controlled Human Malaria Infection; * accuracy except otherwise mentioned).

Vaccine	Vaccinees	Predicted Responses	Predictors	Machine Learning Method	Performance *	Reference
Yellow fever vaccine (YF-17D)	Healthy adults	The magnitude of the activated CD8+ T cell and neutralizing Ab responses	Early blood transcriptional signatures	ClaNC and DAMIP	Up to 90% and 100% respectively	[52]
Seasonal Trivalent Inactivated influenza Vaccine (TIV)	Patients 50–89 years old suffering from multiple chronic medical conditions	The magnitude of plasma HAI Ab response	Baseline signatures among 26 input continuous or categorical variables inc. previous vaccination, low grade chronic inflammation, chronic infections, blood cell counts	Neural network (multilayer perceptron (MLP), radial-basis function network (RBFN) and probabilistic network (PNN)) and Logistic regression	72.5% of average hit rate across 10 samples	[184]
Seasonal Trivalent Inactivated influenza Vaccine (TIV)	Healthy adults	The magnitude of plasma HAI Ab response	Early blood transcriptional signatures	DAMIP	Up to 90%	[185]
Seasonal Trivalent Inactivated influenza Vaccine (TIV)	Healthy adults, inc. young (20–30 years) and older subjects (60 to 89 years)	The magnitude of plasma HAI Ab response	Baseline blood transcriptional, cytokines and cell populations signatures	Logistic regression	84%	[178]
Seasonal Trivalent Inactivated influenza Vaccine (TIV) and pandemic H1N1 (pH1N1) vaccine	Healthy adults	The magnitude of the Ab response	Baseline HAI titer, blood cell populations, transcripts and pathways signatures	Diagonal linear discriminant analysis (for cell frequency data and when cell frequency and pathway status were combined); or partial least square (for data dimension reduction due to the large number of genes) followed by linear discriminant analysis (PLS-LDA) for transcript data alone	0.86 of AUROC	[60]
Seasonal Trivalent Inactivated influenza Vaccine (TIV) over 5 seasons	Human adults, inc. elderlies (>65 years)	The magnitude of plasma HAI Ab response	Early blood transcriptional signatures	DAMIP and artificial neural network classifier	>80%	[10]
Seasonal Trivalent Inactivated influenza Vaccine (TIV)	Healthy adults (50 to 74 years)	The magnitude of the B-cell ELISPOT and plasma HAI Ab responses	Early blood cell composition, mRNA-Seq, and DNA methylation signatures	The ensemble learner (inc. Generalized linear models, Recursive Partitioning, and Regression Trees), and random forest models	0.64–0.79 of AUROC	[186]
Seasonal Trivalent Inactivated influenza Vaccine (TIV)	Healthy adults	The magnitude of plasma HAI Ab response	Baseline HAI titer and blood transcriptional signatures	Gaussian Mixture Model (GMM)	R^2^ = 0.64 for the correlation between observed and predicted data	[187]
Seasonal Trivalent Inactivated influenza Vaccine (TIV)	Healthy adults	The magnitude of the Ab response	Early blood transcriptional signatures	Logistic Multiple Network-constrained Regression	69%	[188]
Seasonal Trivalent Inactivated influenza Vaccine (TIV) over 8 seasons	Healthy adults	The magnitude of the specific Ab response	Baseline blood cell populations signatures	128 machine learning algorithms suitable for classification using Sequential Iterative Modeling “OverNight” (SIMON), inc. Diagonal Discriminant Analysis, Partial Least Squares, Linear Discriminant Analysis, Logic Regression, Neural Network, Random Forest	Up to 0.92 of AUROC	[179]
Seasonal Trivalent Inactivated influenza Vaccine (TIV) given transcutaneously, intradermally or intramuscularly	Healthy adults	The magnitude of the specific T CD8+ and Ab responses	Early blood transcriptional and serum cytokines signatures	Logistic regression	0.93 to 0.96 of AUROC	[189]
Seasonal Trivalent Inactivated influenza Vaccine (TIV) and 23-valent pneumococcal polysaccharide vaccine	Old patients (>65 years) with chronic kidney disease with or without non-dialysis	The magnitude of the HAI Ab and anti-PnPS IgG responses	Baseline signatures among 30 input continuous or categorical variables inc. previous vaccinations, low grade chronic inflammation, chronic infections, blood cell counts	Multivariable linear regression model	*p* < 0.05	[190]
RTS,S malaria vaccine	Healthy adults	The protection against CHMI	Early blood transcriptional signatures	DAMIP	>80%	[181]
Candidate malaria vaccine composed of a Self-Assembling Protein Nanoparticles presenting the malarial circumsporozoite protein (CSP) adjuvanted with three different liposomal formulations: liposome plus Alum, liposome plus QS21, or both	Rhesus macaques	Adjuvant condition	Vaccine-induced immune response signatures among many variables inc. serology, fluorospot, ICS from blood, liver, LN and spleen	Random forest followed by Linear regression analysis	92%	[32]
Live-attenuated varicella zoster virus (VZV) vaccine	Healthy adults, inc. younger (25–40 years) and older (60–79 years)	The magnitude of the specific T and IgG responses	Early blood transcriptional, metabolite clusters, cytokines, and cell populations signatures	Multivariate regression model (Partial least square)	*p* < 0.05	[180]
Monovalent oral polio vaccine type 3 (mOPV3)	Infants aged 6–11 months	Seroconversion or shedding of vaccine virus as a marker of vaccine “take”	Baseline enteric pathogens blood cell populations, and plasma cytokines signatures	Random forest	58%	[191]
Two distinct live attenuated Tularemia vaccine administered by scarification	Healthy humans	The magnitude of the specific Ab and activated CD4 and CD8 T cell responses	Early blood transcriptional signatures	Logistic regression	26% of mean misclassification error	[39]
rVSV-ZEBOV	Healthy adults	The magnitude of the Ab response	Early blood transcriptional, plasma cytokine and cell populations signatures	Sparse partial least-squares followed by multivariable linear regression	0.77 of root square residuals leave-one-out explaining 55% of the variability	[12]
DNA/rAd5 HIV-1 preventive candidate vaccine	Healthy adults	HIV infection	Magnitude and quality of CD4 and CD8 T cells	PCA followed by Cox proportional hazards regression model, and Logistic regression with lasso	Up to 0.75 of AUROC	[192]
Seven preventive HIV-1 vaccine regimens (inc. DNA, NYVAC, ALVAC, MVA, AIDSVAX)	Healthy adults	The magnitude of long-term immune responses	Baseline demographic variables and peak immune responses	Regularized random forest and linear regression models	R = 0.91 for the correlation between observed andpredicted data	[193]
41 different vaccine vectors all expressing the same antigen	Mice	The quality of late T-cell responses	Early transcriptome of dendritic cells	Random forest	Up to 98%	[194]

## Data Availability

Not applicable.

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
