# Peer review of "Predictive Markers of Immunogenicity and Efficacy for Human Vaccines"

_vaccines, 2021, doi:10.3390/vaccines9060579_

Round 1
Reviewer 1 Report
In the manuscript titled “Predictive markers of immunogenicity in animal models of human vaccines”, the authors reviewed several high dimensional experimental techniques and bioinformatics methods serving as promising predictors to assess the vaccine efficacy. Those methods cover the immune responses at different levels which can be integrated as the systematic vaccine “signatures”. In general, the review is well written, comprehensive and easy to follow; there are some concerns which should be addressed before it is accepted as the current form.
1. The power of the adaptive immune system relied on the immune repertoire comprised of diverse genetic population of T and B cells. Therefore, the capacity to accurately quantify the genetic arranges in immune repertoire will play an important role in vaccine research and design. The immune repertoire sequencing has being applied for this purpose with some achievements (https://www.cell.com/trends/immunology/pdf/S1471-4906(14)00077-5.pdf). In this manuscript, the repertoire sequencing was not discussed sufficiently as the technique is only mentioned in the section of difficulties for single cell sequencing without any examples of its practical application.
2. In section 5.2, the introduction of some ML methods and their prime principles can be rephrased in a more concise way, since they are not the focus of this manuscript. There could a table to compare the advantage and disadvantage, as well as the scope of application, of different ML methods.
3. In table 1, it will be more interesting if one column can be added to show the accuracy/efficacy of the predictors and the ML methods from each study.
Minor points:
1. Figure 1, under the context of manuscript, the vaccine efficacy is determined by all factors listed; but the figure gives the readers an impression that there is a linear relationship. In addition, the meaning of “breath” and “persistence” are not clear and accurate in the figure, as the evaluation of a vaccine efficacy relies more on the protection result.
2. Figure 2, “host microbiome” can be added as another factor at the individual level, which can also contribute to modulating the immune response.
3. The limitation of translating strong immunogenicity into efficient protection for vaccine development can also be discussed in the section of challenges, which has been shown in the development of many vaccines against HIV infection.
Author Response
Response to the reviewers:Reviewer #1
Major points: 1. The power of the adaptive immune system relied on the immune repertoire comprised of diverse genetic population of T and B cells. Therefore, the capacity to accurately quantify the genetic arranges in immune repertoire will play an important role in vaccine research and design. The immune repertoire sequencing has being applied for this purpose with some achievements (https://www.cell.com/trends/immunology/pdf/S1471-4906(14)00077-5.pdf). In this manuscript, the repertoire sequencing was not discussed sufficiently as the technique is only mentioned in the section of difficulties for single cell sequencing without any examples of its practical application.
We agree with the reviewer. We have added a short paragraph on immune repertoire sequencing insection 3.3.2.
In section 5.2, the introduction of some ML methods and their prime principles can be rephrased in a more concise way, sincethey are not the focus of this manuscript. There could a table to compare the advantage and disadvantage, as well as the scope of application, of different ML methods.
We do believe that ML methods are not the focus of this review. However, ML methods are essential to go beyondthe correlation analysis between early markers and later responses, and to identify predictors of vaccine immunogenicity or efficacy. Despite being key, it is admittedly complex for most vaccinologists with no background in computational biology to understand the advantages and disadvantages of each method without a bit of technical details.
This being said, we agree that the section on the ML methods could stand as a table instead of a text,thus we have shortened the second paragraph (deletion of lines 769-816) and included an additional table comparing the main principle, and the pros and cons of each ML methods.
3. In table 1, it will be more interesting if one column can be added to show the accuracy/efficacy of the predictorsand the ML methods from each study.
The reviewer is correct and an additional column was added to the table to show the performance of the models, although there are different strategies to assess a model performance, which also depend on the ML method (such as correlation coefficients between the observed and predicted responses, average hit rate across samples, or root square residuals leave one out).
Minor points:
1.Figure 1, under the context of manuscript, the vaccine efficacy is determined by all factors listed; but the figure gives the readers an impression that there is a linear relationship. In addition, the meaning of “breath” and “persistence” are not clear and accurate in the figure, as the evaluation of a vaccine efficacy reliesmore on the protection result.
We agree, we have modified the configuration of Figure 1 to avoid the impression of linear relationship and we have simplified the terms to “vaccine efficacy and safety”.
2. Figure 2, “host microbiome” can be added as another factor at the individual level, which can also contribute to modulating the immune response.
Figure 2 and its caption have been modified accordingly.
3. The limitation of translating strong immunogenicity into efficient protection for vaccine development can also be discussedin the section of challenges, which has been shown in the development of many vaccines against HIV infection.
A small paragraph has been addedto discuss this point insection7. Conclusions
Reviewer 2 Report
This is a timely, comprehensive and critical review on novel methods to predict the efficacy and safety of vaccines. Figures are of good quality. Table 1 is original and useful as a guide. Over 250 well selected references. The MS is well organized and well written. Statistical methods and artificial intelligence are adequately addressed.
In my opinion, the TITLE could be changed to something like: Predictive markers of immunogenicity and safety for human vaccines. Animal models are correctly addressed, but a large part of the MS is addressing important ways to assess the response of humans in blood and other specimens.
To better serve the reader, two short sections should be added: 1) the problem of PHARMACOGENOMICS (also in relation to different ethnic groups worldwide), 2) CONCLUSIONS, that is Authors should express their opinion on which test procedures are likely to be implemented in the short-term and to provide benefits in the quest for novel vaccines.
In addition: The Imaging section may be shortened. Line 833: developments. Start a new paragraph: Table 1.
Abstract and Introduction: reference is made to Zika virus and SARS-CoV-2. However, other viral agents represent important priorities for vaccines. For instance, Dengue and Yellow fever. Reference could be also added to tuberculosis and malaria. Table 1 refers multiple times to traditional influenza vaccines that are among the least effective remedies. Even if examples and pressures abound in the field, emphasis on influenza should be avoided until novel “paninfluenza vaccines” will be made real.
Author Response
1.In my opinion, the TITLE could be changed to something like: Predictive markers of immunogenicity and safety for human vaccines. Animal models are correctly addressed, but a large part of the MS is addressing important ways to assess the response of humans in blood and other specimens.
The title has been changed to “Predictive markers of immunogenicity and efficacy for human vaccines”. Vaccines safety is not the main focus of our review so we did not include safetyin the title though we mention it in the revised version of Figure 1.
2. To better serve the reader, two short sections should be added: 1) the problem of PHARMACOGENOMICS (also in relation to different ethnic groups worldwide), 2) CONCLUSIONS, that is Authors should express their opinion on which test procedures are likely to be implemented in the short-term and to provide benefits in the quest for novel vaccines.
We agree with the reviewer. Two short paragraphs have been added to address the problem of pharmacogenomics: in section 2 and in section 6.2.Besides, we have modified the section 7 (previously “Perspectives and challenges”) to “Conclusions” and enriched this section according to the reviewer’s comment.
3. In addition: The Imaging section may be shortened.
This section has been shortened, though not drastically as there are many technologies and applications worth mentioning for the study of vaccine responses.
4. Line 833: developments. Start a new paragraph: Table 1.This has been corrected.
5. Abstract and Introduction: reference is made to Zika virus and SARS-CoV-2. However, other viral agents represent important priorities for vaccines. For instance, Dengue and Yellow fever. Reference could be also added to tuberculosis and malaria.
A paragraph, with references, has been addedto address this pointin the section 1. Introduction.
6. Table 1 refers multiple times to traditional influenza vaccines that are among the least effective remedies. Even if examples and pressures abound in the field, emphasis on influenza should be avoided until novel “paninfluenza vaccines” will be made real.
As a matter of fact, many models use data from clinical trials evaluating the immunogenicity of traditional flu vaccines, despite their well-known limits (includinglow immunogenicity in elderlies, not pan-influenza viruses vaccine). There might be several reasons for that, including practical reasons (annual immunizations of adults with a safe vaccine) and because the immune correlate of protection are known. We have decided to keep the examples of predictions of antibodyresponses to the current flu vaccines because the methods are valid, but we tempered our comments by acknowledging their limits (in the paragraph following the table).
Round 2
Reviewer 1 Report
The manuscript of "Predictive markers of immunogenicity and efficacy for human vaccines" has been greatly improved after the revision. The manuscript now provides a comprehensive overview of the wide range of applications on immunogenicity predictions for human vaccines.